Eternal-MAML: a meta-learning framework for cross-domain defect recognition

Feng Jipeng 1 2
Zhang Haigang zhg2018@sina.com 2
Wang Zhifeng 1
1 School of Computer Science and Software Engineering, University of Science and Technology Liaoning , Anshan , China
2 Institute of Applied Artificial Intelligence of the Guangdong-Hong Kong-Macao Greater Bay Area, Shenzhen Polytechnic University , Shenzhen , China
Kong Xiangjie
Electronic publication date: 2025 May 7
Publication date: 2025
Volume: 11
Electronic Location ID: e2757
Received 2024 Oct 17; Accepted 2025 Feb 19
Copyright: ©2025 Feng et al.
Copyright year: 2025
Copyright holder: Feng et al.
License: This is an open access article distributed under the terms of the Creative Commons Attribution License, which permits unrestricted use, distribution, reproduction and adaptation in any medium and for any purpose provided that it is properly attributed. For attribution, the original author(s), title, publication source (PeerJ Computer Science) and either DOI or URL of the article must be cited.
License URL: https://creativecommons.org/licenses/by/4.0/

Keywords: Computer vision, Model-agnostic meta-learning, Industrial visual detection

Funding: The Research Projects of Department of Education of Guangdong Province 2023ZDZX1081 2023KCXTD077 University-Enterprise Joint Research and Development Center 602431007PQ This work was supported by the Research Projects of Department of Education of Guangdong Province (2023ZDZX1081, 2023KCXTD077), and University-Enterprise Joint Research and Development Center (602431007PQ). The funders had no role in study design, data collection and analysis, decision to publish, or preparation of the manuscript.

==============================
Defect recognition tasks for industrial product suffer from a serious lack of samples, greatly limiting the generalizability of deep learning models. Addressing the imbalance of defective samples often involves leveraging pre-trained models for transfer learning. However, when these models, pre-trained on natural image datasets, are transferred to pixel-level defect recognition tasks, they frequently suffer from overfitting due to data scarcity. Furthermore, significant variations in the morphology, texture, and underlying causes of defects across different industrial products often lead to a degradation in performance, or even complete failure, when directly transferring a defect classification model trained on one type of product to another. The Model-Agnostic Meta-Learning (MAML) framework can learn a general representation of defects from multiple industrial defect recognition tasks and build a foundational model. Despite lacking sufficient training data, the MAML framework can still achieve effective knowledge transfer among cross-domain tasks. We noticed there exists serious label arrangement issues in MAML because of the random selection of recognition tasks, which seriously affects the performance of MAML model during both training and testing phase. This article proposes a novel MAML framework, termed as Eternal-MAML, which guides the update of the classifier module by learning a meta-vector that shares commonality across batch tasks in the inner loop, and addresses the overfitting phenomenon caused by label arrangement issues in testing phase for vanilla MAML. Additionally, the feature extractor in this framework combines the advantages of the Squeeze-and-Excitation module and Residual block to enhance training stability and improve the generalization accuracy of model transfer with the learned initialization parameters. In the simulation experiments, several datasets are applied to verified the cross-domain meta-learning performance of the proposed Eternal-MAML framework. The experimental results show that the proposed framework outperforms the state-of-the-art baselines in terms of average normalized accuracy. Finally, the ablation studies are conducted to examine how the primary components of the framework affect its overall performance. Code is available at https://github.com/zhg-SZPT/Eternal-MAML.

Introduction

In manufacturing, inspection of defects in the appearance of industrial products is a crucial production process. Quickly distinguishing product defects helps manufacturers obtain information quickly about the production process and make necessary adjustments, thereby reducing the cost waste resulting from untimely inspections. Traditional inspection of industrial defects is performed manually, but as inspection time increases, quality inspectors may experience fatigue of the eye, leading to missed or incorrect inspections of industrial products. Currently, computer vision technology dominates the field of industrial product defect detection (Bai et al., 2024). Generally, industrial defect visual detection systems consist of four parts: image acquisition, intelligent inspection algorithms, data communication, and decision-making and execution. Among them, intelligent inspection algorithms are the core. Relying on traditional image processing techniques (Khanam et al., 2024) for industrial defect recognition has a long-standing research history. One important method involves recognizing defects based on the target shapes by manually designing specific feature extractors to extract pixel-level and structure-level image features. These extracted features are then fed into traditional classifiers such as k-neural networks (KNN) (Guo et al., 2003) and support vector machines (SVM) (Hearst et al., 1998) for defect recognition. However, two main factors limit the application of traditional image processing techniques in the visual detection of industrial product defects. Firstly, manual feature extraction has significant limitations, as it cannot guarantee the model’s ability to accurately characterize defects. Secondly, traditional image processing algorithms are task-driven and severely lack model generalization.

Deep neural networks (LeCun, Bengio & Hinton, 2015) have propelled the advancement of computer vision technology, showcasing remarkable performance in industrial defect visual detection tasks. Typically, industrial defect visual detection exhibits unique characteristics that necessitate targeted modifications to deep neural network models. Firstly, there is the issue of detecting small targets. In industrial product defect detection, the defective areas often occupy a small proportion of the overall sample. Taking the HRIPCB dataset (Huang et al., 2020) as an example, in extreme cases, the area of the defect may be as little as one two-hundred-thousandth of the total sample area. Secondly, industrial products come in a wide variety, and defect characteristics can be influenced by complex background interferences. In such cases, more powerful deep neural networks (He et al., 2016; Han et al., 2020; Abbas et al., 2024) are necessary. Finally, and most importantly, industrial defect detection tasks often face serious issues with few-shot and zero-shot defect scenarios. The most direct and effective method for handling imbalanced samples is transfer learning, which involves applying pretrained models to industrial defect detection tasks through fine-tuning.

Transfer learning (Yan et al., 2024) necessitates addressing domain shift issues in industrial visual detection tasks, which arise from varying feature distributions across different industrial defect detection datasets. Mandatory knowledge transfer not only fail to effectively guide model training but also lead the model astray. Traditional transfer learning tends to be more effective for natural datasets because the mutually exclusive characteristics between different domains in natural datasets are more pronounced. By merely fine-tuning pre-trained initialization parameters on a new domain dataset, the model can quickly generalize. Unfortunately, industrial product defects often manifest at the pixel level, and the shape of the same defect can vary significantly across different industrial products. This variation poses a substantial challenge for traditional transfer learning, making it difficult to transfer knowledge effectively from one domain to another.

Data augmentation and auxiliary information support are effective methods for addressing the challenges of few-shot learning. Multi-task learning (Hu et al., 2024) is a strategy that can mitigate data scarcity by learning useful features and performing accurate classification with only a small amount of labeled data. Frameworks proposed by Xie et al. (2022) and Jain et al. (2022) significantly improve the accuracy and robustness of surface defect recognition by integrating auxiliary information and attention mechanisms. Additionally, using generative models for data augmentation (Catak et al., 2021) effectively addresses the issue of insufficient training data. However, these two methods are inadequate for tackling few-shot and zero-shot industrial defect visual inspection tasks (Zeng & Zubiaga, 2022). On one hand, auxiliary information, similar to defect data samples, is difficult to obtain; on the other hand, data augmentation through generative strategies lacks sufficient diversity, resulting in inadequate model generalization.

Handling pixel-level defect types and adapting to cross-domain issues can be highly effective with meta-learning. Model-Agnostic Meta-Learning (MAML) (Finn, Abbeel & Levine, 2017) aims to provide a general meta-initialization parameter, allowing the model to quickly adapt to new tasks through a few gradient updates after loading the parameters. MAML effectively addresses cross-domain defect recognition tasks by learning a robust parameter initialization, enabling the model to quickly adapt to new tasks even with minimal labeled data. This capability is particularly suitable for cross-domain defect recognition, as defect patterns can vary significantly across domains and labeled data is often scarce. Several meta-learning frameworks, including the Reptile (Nichol, 2018), Almost No Inner Loop (ANIL) (Raghu et al., 2019), and UNICORN-MAML (Ye & Chao, 2021) have been developed. Our previous work introduced the MeDetection (Zhang et al., 2023) framework, which combines MAML and Siamese networks (Chopra, Hadsell & LeCun, 2005) and performs well in defect detection tasks. However, in dealing with industrial visual inspection tasks, MAML has some drawbacks. Firstly, in terms of neural network architecture, it utilizes the classic Conv4 structure, which can lead to gradient vanishing or exploding problems during training. Secondly, due to the nature of MAML’s training strategy, there can be a label shuffling issue, where the same task may have different label assignments. This results in varying generalization accuracy during the testing phase, even for the same task, due to different assigned ground truths.

This article introduces a novel meta-learning framework, termed Eternal-MAML, based on MAML for industrial visual inspection tasks. It aims to address the challenges of few-shot samples in training deep learning models for defect recognition across multiple domains. The proposed Eternal-MAML framework incorporates residual connections and a squeeze-and-excitation module (Hu, Shen & Sun, 2018) into the backbone to address the gradient issues that arise during training with vanilla MAML. Additionally, to avoid the label permutation problem, the framework introduces a new training strategy that learns a one-dimensional vector in the outer loop to guide the update of the classification head in the inner loop. This strategy improves the generalization performance of the learned meta-initialization parameters. Simulation results based on the MVTec benchmark dataset (Bergmann et al., 2019) have demonstrated the effectiveness of Eternal-MAML model. Compared to mainstream advanced algorithms, the proposed framework demonstrates superior performance by fast converging the highest generalization accuracy during testing phase.

Related Works

Known defect patterns

For known defects, supervised learning algorithms are generally used. Traditional methods primarily rely on defect features such as shape, using image processing techniques or a combination of machine learning methods for defect recognition. Edge detection algorithms, like Sobel, Canny, and Prewitt, effectively locate defect areas by extracting edge features from images. Mery & Arteta (2017) combined a simple local binary pattern (LBP) descriptor with SVM linear classifier achieves high precision and accuracy when inspecting metal components using X-ray testing. Zaghdoudi, Bouguettaya & Boudiaf (2024) proposed an improved multi-block LBP algorithm to efficiently detect of surface defects on steel materials. However, these methods have difficulty dealing with complex backgrounds or defects with low signal-to-noise ratios, and they also demand high imaging conditions. CNN, with its powerful data processing and pattern recognition capabilities, can automatically learn differentiated features, significantly enhancing decision-making and prediction accuracy. He et al. (2016) introduced residual connections in neural network architecture, addressing the problem of vanishing or exploding gradients in deep networks. This significantly enhances the ability to automatically classify different types of defects in the same industrial product. Han et al. (2020) introduced GhostNet, which significantly reduces the computational load by generating more feature maps with fewer parameters and computations. This innovation greatly facilitates the implementation of the algorithm in real production environments.

Unknown defect patterns

In real-world situations, previously unseen defects may occur. This requires the use of unsupervised learning (Cebeci et al., 2022). Unsupervised learning is a machine learning method primarily used to discover hidden patterns or features in datasets without labeled data. Unlike supervised learning, unsupervised learning does not rely on known output labels but instead analyzes and processes input data directly. The reconstruction-based approaches are mainstream and have achieved significant results in defect detection. Yong et al. (2023) investigated the use of the CLIP model for building defect detection, leveraging its zero-shot and few-shot learning capabilities. The findings reveal that prompt characteristics significantly affect model performance. Based on this, an optimal prompting strategy was formulated, leading to a detection performance that exceeds that of supervised models. Li, Guo & Wang (2024) proposed improved CycleGAN models to enhance automatic crack detection in bridges by addressing the challenges of manual annotation workload and dataset collection, demonstrating significant improvements in data augmentation and segmentation performance. Additionally, the recently popular diffusion models possess exceptional image reconstruction capabilities, which have inspired researchers to utilize them for enhancing the reconstruction of anomalous images. DiAD (He et al., 2024) combined a pixel-space autoencoder, a latent-space semantic-guided (SG) network, and a feature-space pre-trained feature extractor for multi-class anomaly detection.

Limited defect labeling

When defect samples are scarce and present a significant imbalance compared to normal samples, using traditional deep learning methods to train on imbalanced data is highly likely to result in overfitting. To address this problem, meta learning (Vettoruzzo et al., 2024) works by training models to learn how to learn. It can quickly adapt to new tasks with minimal data by leveraging prior experience from related tasks. There are many useful methods based on meta learning to resolve cross-domain defect recognition. For example, relation network (Sung et al., 2018) is to model and measure the relationships between pairs of entities or features, enhancing the understanding of their interactions. MAML (Finn, Abbeel & Levine, 2017) improves the generalization ability of machine learning models on new tasks by learning initial model parameters, whose characteristic is also rapid generalization. However, MAML has some drawbacks during training, such as instability and low generalization performance during testing. Thus, based on the effectiveness of MAML, improved meta-learning frameworks such as ANIL (Raghu et al., 2019), and UNICORN-MAML (Ye & Chao, 2021) addressed issues like training instability and insufficient generalization performance in complex tasks that occur during the training process of vanilla MAML. However, these improved MAML-based frameworks, while performing well on natural datasets (such as miniImageNet (Vinyals et al., 2016)), do not necessarily exhibit high generalization performance in industrial defect recognition tasks. Primarily because the characteristics and data distributions of natural images differ from those of industrial defect images. Natural images often focus on diverse backgrounds and objects, whereas industrial defect detection requires capturing subtle defect features such as scratches and dents. Therefore, pretrained models struggle to directly adapt to these nuances and specific task requirements.

Method

Problem defination

The cross-domain industrial defect recognition problem is defined as follows: Let D = {D1, D2, …, Dm} represent a set of industrial product datasets, where Di (i = 1, 2...m) represents a specific industrial product dataset. Tij is a task extracted from Di, which is used to distinguish n different defects in Di. Each task contains a support set Sj and a query set Qj. The support set Sj contains (n-way×k-shot) labeled data, while the query set Qj contains (n-way×k-query) unlabeled data. Here, n-way indicates that n different defect categories are randomly selected from Di, k-shot and k-query represent the number of samples per defect category in the support set and query set, respectively. In the same task, the number of samples in the support set and query set is the same, n-way. Usually, k-shot is smaller than k-query, but even if this requirement is not met, it does not affect the effectiveness of our proposed framework. Therefore, in this work, the goal is to use Dtrain = {D1, D2, …, Dk} to train an n-way classifier f, and obtain the well-trained initial parameters. Then, for a completely new industrial product Dt, it only needs to update the parameters in the inner loop through a few steps. The updated parameters can quickly adapt to the defect recognition task of Dt in the outer loop. Here, Dt is randomly selected from Dtest = {Dk+1, Dk+2, …, Dm} for testing industrial products, with Dtrain∪Dtest = D.

MAML in cross-domain defect classification

MAML (Finn, Abbeel & Levine, 2017) is a meta-learning algorithm suitable for any model trainable via gradient descent, and it is effectively used in cross-domain defect recognition due to its ability to generalize quickly with minimal data and gradient updates. In cross-domain defect classification tasks, MAML involves two stages of training:

Inner loop stage

This level of training involves rapid adaptation through a few gradient updates using a small amount of data for a specific task that classifies different defects in one industrial product. Let fθb denote the base learner for a specific task, where θb is the parameter of recognition model. During the training phase, MAML randomly selects a task Tb from a specific industrial product dataset Db, which follows the task distribution P(T). The method of selecting a batch size of tasks each time remains consistent. At this step, the MAML model applies the support set Dbs of Tb to perform i inner-loop updates, where i is the number of inner-loop update steps. The parameter update of the base learner can be expressed as: (1) θib=θi−1b−α∇θi−1bLDbsfθi−1b

where α is the inner-loop learning rate, and LDbsfθi−1b is the loss of the base learner with parameters θi−1b on the support set Dbs after the (i − 1)th step (the same logic as the previous steps).

Outer loop stage

The goal of the outer-loop stage is to evaluate the generalization performance of the initialized parameters obtained after updating the base learner in training tasks and find a robust meta parameter θ. This ensures that a new defect recognition task in a new industrial product can be quickly adapted with only a few gradient updates. By aggregating the performance of inner-loop updates over multiple tasks, the MAML model collects the loss values evaluated on the query sets of the training tasks after the final gradient update in the inner loop for a batch of tasks, and then updates the meta parameters of the model. Specifically, let B be a set of training tasks. For each task in B, after performing the inner-loop updates, the loss on the corresponding query set Dbq is aggregated as follows: (2) Lθ= ∑b=1BLDbqfθNbθ

where θNb represents the parameters of the base learner obtained after the last update of the inner loop, and the meta-learner updates the meta parameters θ based on this loss, that is: (3) θ=θ−β∇θ ∑b=1BLDbqfθNbθ

where β is the learning rate for updating the meta-learner, and the final obtained θ possesses a commonality among the tasks, making it easy to adapt to defect classification tasks in a new industrial product.

Eternal-MAML framework

MAML has distinguished itself in few-shot classification due to its rapid generalization ability, particularly on natural image datasets. Raghu et al. (2019) provides substantial experimental evidence suggesting that MAML’s effectiveness stems from the reuse of features extracted by its feature extractor, the choice of this extractor is therefore crucial within the MAML framework. The original MAML implementation utilizes a four-layer convolutional network, commonly referred to as Conv4, which outperforms traditional transfer learning approaches when classifying natural images. However, a critical distinction arises when considering natural images versus industrial defect datasets. Natural image datasets typically exhibit significant inter-class variations, allowing Conv4 to readily extract discriminative features for accurate classification. In contrast, industrial defect classification often involves subtle and varied defects within the same product type, with some defects manifesting at the pixel level. Consequently, the feature extractor must be capable of capturing fine-grained features to effectively distinguish between different defect types within the same product category. In this context, the limited depth of a four-layer convolutional network may prove inadequate for the intricacies of industrial defect recognition. The limitations of Conv4, particularly its restricted capacity for feature extraction, will be thoroughly investigated in the following experimental simulations focused on defect recognition.

To extract pixel-level defect features for effective defect classification, we incorporate residual blocks and Squeeze-and-Excitation (SE) modules. Figure 1 demonstrates the combined use of SE and residual modules. In the residual block, these are “shortcut” connections in neural networks that bypass some layers, adding the input of a layer to its output. This helps to alleviate the vanishing gradient problem, making it easier to train very deep networks. SE blocks enhance channel-wise feature representation by learning to weight the importance of each channel in a convolutional layer. They “squeeze” global spatial information into a channel descriptor and then “excite” to recalibrate channel-wise responses, effectively boosting informative features and suppressing less useful ones. The integration of residual connections and Squeeze-and-Excitation modules effectively addresses the vanishing or exploding gradient issue while simultaneously enabling the modeling of feature channels, thus facilitating the extraction of discriminative features for distinguishing between mutually exclusive defects. These capabilities contribute to more stable MAML training.

Figure 1 Schematic diagram of the integration of Squeeze-and-Excitation and residual modules.

When applying MAML, augmented with residual connections and Squeeze-and-Excitation modules, to cross-domain defect recognition tasks, we observed only marginal improvements compared to the original MAML. After extensive experimentation and a review of related research on MAML enhancements, we drew inspiration from Ye & Chao (2021). This led us to identify another potential reason for MAML’s suboptimal performance in defect recognition: Inconsistency in the label space caused by random label assignment. Then, what is meant by label permutation randomness? In n-way k-shot classification tasks, MAML aims to learn a good model initialization that enables rapid adaptation to new tasks. However, since each task is independently and randomly sampled, even the same class can be assigned different labels across different tasks. As illustrated in Fig. 2A, when using MAML to distinguish N different defects in hazelnuts with random label permutations, the “crack” and “print” defects are assigned different true labels in different tasks, leading to inconsistencies between the inner and outer loops.

Figure 2 The challenge of random label permutations posed by the original MAML framework.

(A) Illustrates MAML’s problem of random label permutation, while (B) shows how Eternal-MAML reduces its impact by using a meta-vector to guide the inner loop’s classification head. Image source credit: Bergmann et al., 2021.

Specifically, during the inner loop, the model adapts quickly based on the random label space of the current task. Because labels are randomly assigned, the model learns to map specific features to the specific labels of the current task, rather than learning the essential features of the class. This causes the optimization direction in the inner loop to be specific to the label space of the current task, instead of a general direction related to the intrinsic characteristics of the class. The outer loop aims to update the model’s initial parameters to facilitate rapid adaptation to new tasks. However, due to the varying label spaces across tasks, the outer loop effectively tries to find an initialization that can adapt quickly to all possible label space permutations. This makes the optimization objective of the outer loop extremely complex and challenging, as the model needs to adapt to N! possible label spaces. The generation of random label spaces forces the model to expend more computational resources and time adapting to different label spaces rather than learning the essential features of the classes, hindering generalization to new, unseen industrial defect classification tasks.

The typical process of classification tasks involves a feature extractor extracting features, followed by the classification of feature vectors via a classification head. We argue that the performance of the classification head is highly sensitive to the label permutation. In the case of the same task but different true label permutations, the classification head tends to compromise all possible label permutation results, thus deviating from the effective reuse of the features extracted by the feature extractor. To address this issue, we propose a learnable meta-vector W, which is a one-dimensional vector with the same dimension as the N-th dimensional vector of the classification head. By capturing the “common” feature representation of each of the N categories, this meta-vector aims to reduce the negative impact of random label permutations on the optimization of both inner and outer loops. The specific implementation process of the meta-vector is as follows: assuming that the feature dimension output by the feature extractor is d, for an n-classification task, the dimension of the classification head weight Wi is usually (n, d). Each row Wi[j, :] (a d-dimensional vector) can be regarded as the weight vector of the jth category. Before the start of each inner loop, we initialize all N d-dimensional vectors of the learner’s classification head with this meta-vector, and then use the support set of the task to perform inner loop optimization. After the inner loop is completed, the learned initialization parameters are evaluated on the query set, the backpropagation gradient is obtained and used to update the initialization parameters to obtain the meta-training parameters. Finally, we separately obtain the gradient evaluated by the outer learner’s classification head on the query set, and use this gradient to update the meta-vector. Algorithm 1 describes the specific process of Eternal-MAML in inter-task learning in detail.

_______________________ Algorithm 1 Eternal-MAML for few-shot classification______________________________ Require: P(T): distribution of tasks Require: Dsb: support set; Dq b: query set Require: α,β: step size hyperparameters Require: W: Meta-vector Require: {ωc}N1: classifier head weights   1:  Randomly initialize meta parameter θ  2:  Randomly initialize W  3:  while not done do  4:       Get batch of tasks Tb ∼ P(T)   5:       {ωc}N1 ← W  6:       for all Tb do do  7:            Get  the  support  set  Dsb   =   {(xab,yab)}Ka=1,   query  set  Dq b   =      {(xab,yab)}Q a=1   8:            Base update θbi ← θbi−1 − α∇θLDs b(fθb i−1)   9:       end for 10:       Meta update θ ← θ − β∇θ ∑B      b=1 LDq b (fθb N (θ)) 11:       update W ← W −β∇ω ∑B       b=1 ∑N               c=1 LDq b (fωb c(ω)) 12:  end while_________________________________________________________________________

Specially, Algorithm 1 demonstrates the pseudocode of the proposed Eternal-MAML model, which is also divided into two stages: the inner loop to train the base-learner and the outer loop to train the meta-learner. Without loss of generality, the base-learner is trained for specific tasks, while the meta-learner encompasses the commonalities of all training tasks. First, the proposed Eternal-MAML model parameterizes the base-learner as follows: θ = {φ, w1, w2, …, wN} where N represents the number of classifier types in a task. Within this framework, a neural network framework composed of a squeeze-and-excitation module and residual connections is utilized as the standard feature extractor. Before starting the inner loop, the proposed Eternal-MAML model applies a vector W to initialize the classifier heads that are denoted as wcc=1N, then wcc=1N are trained through multiple steps on the task set, and the parameters of base-learner are updated, which adheres to (1), and are evaluated on the query set Dbq to obtain the meta loss, as shown in (2). The goal of the proposed framework is to minimize this loss and update the meta-learner parameters θ through the gradient of this loss, as indicated in (3). Additionally, while updating θ, the model also updates the vector W. The strategy is summarized as follows: (4) W=W−β∇ω ∑b=1B ∑c=1NLDbqfwcbω

where β is the meta-learning rate, the same as the learning rate for updating the meta parameters θ, and wcb represents the weight vectors of different classifier heads in B tasks during inner loop.

Figure 2B visually demonstrates the process of initializing the classification head weights with the meta-vector, thereby circumventing the limitation of poor generalization performance of the learned initialization parameters due to the randomness of label permutations in the classification head. Moreover, Fig. 3 illustrates the complete architecture of Eternal-MAML. We maintain the core training strategy of the original MAML, and the integration of the two key modifications discussed above contributes to the effectiveness of our framework in the context of cross-domain defect classification. However, the incorporation of residual and squeeze-and-excitation blocks, along with the introduction of the meta-vector, results in a higher training cost for Eternal-MAML compared to the original MAML.

Figure 3 Eternal-MAML model, retaining the core framework of MAML, the introduced improvement strategies demonstrate superior performance in cross-domain defect recognition tasks.

Image source credit: Bergmann et al., 2021.

Experiments and Results

Experimental setup

In this study, we used PyTorch with a graphics processing unit (GPU) to build the Eternal-MAML framework. The details of the hardware and software configurations are shown in Table 1. To learn an n-way classifier for cross-domain defect recognition, n is set to 2 or 3, k ∈ {1, 3, 5}, and k-query is fixed at 10. The model was trained with an inner-loop learning rate of 1e−2, an outer-loop learning rate of 1e−3, the inner loop optimization was performed for five steps and images resized to 200 × 200. Adam optimizer is chosen with a learning rate of 1e−3 and a weight decay of 5e−2. The network structure settings include a drop rate of 0.1 and drop block size of 2. Training uses 100 epochs with a batch size of 16 tasks. During the testing phase, three mutually exclusive domains of industrial products are randomly selected from the test set to evaluate the generalization performance of the trained meta parameters. For each of these three industrial products, 50 tasks are randomly selected, with each task involving distinguishing between n different defect types of the same industrial product.

Table 1 Experimental setup.

Item	Type	
CPU	Intel(R) Xeon(R) W-2255 CPU @ 3.70 GHz 3.70 GHz	
RAM	32.0GB	
GPU	NVIDIA GeForce RTX 3080	
Operating system	Linux	
CUDA	CUDA 11.8	
Programming language	Python 3.9	
Deep learning frame	PyTorch 2.2.1	

Experimental data

The performance of the proposed Eternal-MAML model is evaluated using the MVTec (Bergmann et al., 2019) and NEU-CLS (Bao et al., 2021) datasets, which are publicly available benchmark datasets for few-shot defect classification and can be accessed through their respective publications. MVTec includes 15 different categories of industrial products, each with normal samples and various types of defective samples. For cross-domain defect recognition using MVTec, the 15 types of industrial product are randomly recombined to form six different folds. In each fold, eight types of industrial products are used as a training set, two types are used as the validation set, and the remaining four types serve as the test set. The validation set is utilized to carefully select hyperparameters to ensure the best results for each experiment. Figure 4 displays normal samples and different types of defect samples for five types of industrial products in one fold. The NEU surface defect database contains images (200 × 200 pixels) of six typical surface defects, which includes 1,800 grayscale images: 300 samples each of these typical surface defects. It presents two main challenges for defect detection: high intra-class variation in appearance (e.g., scratches can be horizontal, vertical, or slanted) and high inter-class similarity (e.g., rolled-in scale, crazing, and pitted surface). Illumination and material changes further complicate defect identification by affecting image grayscale. Given that the NEU-CLS dataset solely contains data representing diverse steel defect categories, it is unnecessary to re-partition it. For each task, n different defect types are directly sampled, and k-shot + k-query images are collected for each chosen defect type.

Figure 4 Randomly selecting five types of industrial products from a certain fold, we extract normal samples and five different types of defective samples for each product.

Some defects can be observed at a pixel level, and the shape and color of the same type of defect can vary significantly across different industrial products. This places high demands on the performance of the model, requiring it to accurately extract mutually exclusive features for precise defect detection and to adapt to different domains. Image source credit: Bergmann et al., 2021.

Results and analysis

To validate the effectiveness of the proposed framework, a comparative analysis was undertaken, considering six established baseline models commonly used in cross-domain few-shot defect classification research. The methodology is outlined below. Training from scratch: Training from scratch refers to the process of training a machine learning model without using any pre-trained weights, starting instead with randomly initialized weights. All model parameters are then adjusted during the training process based on the data and the backbone is the same as Eternal-MAML. ResNet-12: It has become a commonly used backbone network in few-shot classification research, achieving excellent results on several popular few-shot learning benchmark datasets, such as miniImageNet. In this experiment, it is employed as the baseline model for a few-shot defect classification task. ResNet-50: With its robust feature extraction capabilities, potential for transfer learning, and strong generalization ability, presents an effective model choice for few-shot defect detection. By pre-training on large-scale datasets and fine-tuning on limited defect samples, ResNet-50 can learn discriminative features and achieve accurate defect detection. MAML (Finn, Abbeel & Levine, 2017): Details of the training process can be found in the second part of METHOD. UNICORN-MAML (Ye & Chao, 2021): This framework, proposed by Ye & Chao (2021) improves MAML by explicitly increasing step size (M>10). Each new training uses a vector-initialized classifier head, but the direction should remain unchanged during the training process, which differs from the proposed Eternal-MAML model. It offers a more effective and stable variant of MAML for few-shot classification tasks. ANIL (Raghu et al., 2019): It is an improved version of MAML, designed to enhance the efficiency and performance of meta-learning. Within the inner loop, only the head parameters are updated, while the feature extractor’s parameters remain frozen. Moreover, an even more extreme approach is to entirely forego updates to the head parameters within the inner loop, modifying solely the feature extractor. This implies that the classifier for each task is initialized using a pre-trained classifier and is kept constant throughout the meta-training process.

Experiments were conducted on the proposed framework and other comparative methods using all folds. These experiments involved training a comprehensive n-way classifier for k (where k ∈{1, 3}). The average normalized accuracy is considered as the metric to evaluate the generalization performance of model. Therefore, we use Accnor to represent it. Accnor is averaged over the normalized accuracy of each defect class within the same industrial product, providing a comprehensive evaluation of the classifier’s performance on a new industrial product defect detection task. The calculation method involves dividing the number of correctly classified samples of each defect class by the total number of samples in that class, and then averaging the normalized accuracy of all classes. Accnor better reflects the model’s classification ability across different industrial products, which is defined as follows: (5) Accnor=Acccls−Accrc1−Accrc

where Acccls represents the accuracy of a specific defect class in one industrial product, usually referring to the proportion of correctly identified samples in a classification task of a specific class. Accrc can be understood as the expected accuracy of a random classifier, for example, in training an n-way classifier, the expected accuracy of a random classifier is 1/n-way. A higher Accnor value indicates better generalization accuracy, while Accnor ≤ 0 implies that the trained n-way classifier does not generalize well.

We conducted a comprehensive evaluation of our proposed method for few-shot defect classification, comparing its performance against several established baselines, including ResNet-12, ResNet-50, training a model from scratch, and meta-learning approaches such as MAML, UNICORN-MAML, and ANIL. Table 2 presents the results of these experiments using a 6-fold cross-validation strategy. Our method achieves an average accuracy of 0.5397 across all folds, which is mainly attributed to the fact that residual connection and Squeeze-and-Excitation block can extract mutually exclusive defect features, while the Meta-vector maximally reduces the impact of label permutation on training and testing by guiding the update of the classification head of the inner-loop learner. Specifically, our method outperforms the second-best performing method, Ours+Conv4 (0.5185), by a margin of 2.12%. This further indicates that applying residual connection and Squeeze-and-Excitation block to MAML improves generalization accuracy by addressing gradient issues and modeling channel attention, leading to better pixel-level defect extraction. Also, our method significantly surpasses ANIL (0.4467) by 9.3%. The results also underscore the challenges of training from limited data, as evidenced by the low performance of the model trained from scratch (0.3034). While ResNet-12 or ResNet-50 provides a reasonable baseline (0.3192/0.4505), it still lags behind the meta-learning approaches and our method. MAML (0.3965) and UNICORN-MAML (0.4586), while improving upon training from scratch, do not achieve the performance levels of our proposed method. Furthermore, the performance across different folds reveals interesting insights. While our method consistently performs well in average, the notably high accuracy achieved in fold 2(0.5034), fold 4(0.5518) and fold 5(0.6323) warrants further investigation. This suggests that certain characteristics of the data within this fold may be particularly advantageous for our method. Analyzing the specific data composition of this folds could provide valuable insights for understanding the strengths and limitations of our approach, potentially leading to further improvements in its performance across diverse datasets and experimental settings. Future work will explore this aspect in more detail.

Table 2 Comparing the proposed framework with other six baseline methods, learning a 2-way classifier accross six folds of Dtest.

Our proposed framework consistently outperforms the rest of the baselines across all folds. Each result is calculated by averaging the performance obtained in 1-shot, 3-shot, and 5-shot experiments. Best results are in bold.

Method	Fold	Average	
	1	2	3	4	5	6	
Training from scratch	0.2423	0.3261	0.3150	0.2412	0.3671	0.3291	0.3034	
ResNet-12	0.2212	0.3923	0.3442	0.2863	0.3211	0.3503	0.3192	
ResNet-50	0.4102	0.3567	0.4033	0.4412	0.6415	0.4502	0.4505	
UNICORN-MAML	0.3807	0.4627	0.4223	0.4718	0.5821	0.4323	0.4586	
ANIL	0.4544	0.5134	0.4877	0.3656	0.4783	0.3810	0.4467	
MAML	0.3411	0.4001	0.4912	0.3434	0.4323	0.3710	0.3965	
Ours+Conv4	0.4409	0.5122	0.4945	0.5534	0.5963	0.5140	0.5185	
Ours+SE-ResNet-12	0.4622	0.5034	0.5445	0.5518	0.6323	0.5445	0.5397	

Table 3 presents a comparative analysis of the Eternal-MAML method against six baseline approaches in the context of learning a three-way classifier on six distinct folds of the MVTec dataset. The results unequivocally demonstrate the superior performance of Eternal-MAML, specifically when implemented with the SE-ResNet-12 architecture (Ours+SE-ResNet-12), which achieves the highest or near-highest accuracy across all six folds, culminating in a leading average accuracy of 0.4660. The proposed method, when used with a simpler architecture, namely Conv4 (Ours+Conv4), exhibits the second-best performance with an average accuracy of 0.4366, further highlighting the effectiveness of the proposed approach. Notably, it achieves the highest accuracy on fold 6 (0.5091). ANIL and UNICORN-MAML demonstrate competitive performance, yielding average accuracies of 0.3950 and 0.3853, respectively. While ANIL outperforms all other methods on fold 2, attaining an accuracy of 0.4241. The transfer learning, namely ResNet-12 and ResNet-50, display moderate performance, with average accuracies of 0.3034 and 0.3584, respectively, suggesting a potential limitation in their generalization capabilities within this specific task. Crucially, the training-from-scratch approach exhibits the weakest performance, achieving an average accuracy of only 0.0422 and, critically, failing completely on fold 2, 4, and 6 with an accuracy of 0. This observation underscores a significant deficiency in its ability to generalize to unseen data within these specific folds. The standard MAML approach also underperforms, with a mean accuracy of only 0.1688.

Table 3 Comparing the Eternal-MAML with six baseline methods, learning a 3-way classifier on six sub-datasets of MVTec dataset.

Under the comparison of comprehensive generalization performance, Eternal-MAML performs the best. Specifically, training from scratch has no generalization ability on some folds. Best results are in bold.

Method	Fold	Average	
	1	2	3	4	5	6	
Training from scratch	0.1211	–	0.0810	–	0.0513	–	0.0422	
ResNet-12	0.2034	0.3877	0.3522	0.2776	0.2214	0.3781	0.3034	
ResNet-50	0.3102	0.3456	0.4133	0.4398	0.3415	0.3002	0.3584	
UNICORN-MAML	0.3823	0.3602	0.3766	0.4793	0.4873	0.2261	0.3853	
ANIL	0.3544	0.4241	0.4012	0.3411	0.4583	0.3910	0.3950	
MAML	0.2864	0.2126	0.1253	0.1493	0.1346	0.1047	0.1688	
Ours+Conv4	0.4809	0.4032	0.4321	0.3512	0.4433	0.5091	0.4366	
Ours+SE-ResNet-12	0.5223	0.3921	0.4432	0.5012	0.5310	0.4062	0.4660	

On the NEU-CLS dataset, we evaluated the performance of Eternal-MAML against current state-of-the-art methods for few-shot defect classification. Table 4 displays experiments encompassed 2-way and 3-way classification tasks, under both 1-shot and 5-shot learning settings. In the 2-way 1-shot setting, Eternal-MAML (SE-ResNet-12) achieved an accuracy of 0.8360, outperforming ANIL (0.8220), UNICORN-MAML (0.7660), and MAML (0.7387), among others. In the 2-way 5-shot setting, Eternal-MAML (SE-ResNet-12) continued to lead with an normalized accuracy of 0.9665, followed closely by ANIL at 0.9708. Under the more challenging 3-way 1-shot setting, Eternal-MAML (SE-ResNet-12) maintained its superior performance with an accuracy of 0.7916, surpassing ANIL (0.7001), UNICORN-MAML (0.6277), and MAML (0.6416). In the 3-way 5-shot setting, Eternal-MAML (SE-ResNet-12) still led (0.8316), followed by Eternal-MAML (Conv4) (0.8003). These results demonstrate that our proposed Eternal-MAML (SE-ResNet-12) exhibits robust generalization capability regardless of task complexity, highlighting its effectiveness in few-shot defect classification tasks. Notably, ANIL’s accuracy surpassed that of Eternal-MAML (SE-ResNet-12) in the 2-way 5-shot setting, which can be attributed to the inherent characteristics of the ANIL algorithm.

Table 4 Performance comparison on NEU-CLS dataset under 2-way and 3-way with 1-shot and 5-shot settings.

Best results are in bold.

Dataset	NEU-CLS	
n-way	2-way	3-way	
Setups	1-Shot	5-Shot	1-Shot	5-Shot	
Training from scratch	0.5221	0.6114	0.4200	0.5667	
ResNet-12	0.6391	0.7266	0.6777	0.6989	
ResNet-50	0.6208	0.7868	0.6316	0.6990	
UNICORN-MAML	0.7660	0.9230	0.6277	0.7767	
ANIL	0.8220	0.9708	0.7001	0.7168	
MAML	0.7387	0.8775	0.6416	0.7516	
Ours (Conv4)	0.7422	0.9244	0.7832	0.8003	
Ours (SE-ResNet-12)	0.8360	0.9665	0.7916	0.8316	

Ablation experiments

In this section, extensive ablation studies were conducted on the main components of the Eternal-MAML framework while learning a 2-way classifier across all folds. We conducted experiments by removing some components of the proposed framework, and compared the results with the complete framework, performing a total of four different experiments. Similarly, we used the average normalized accuracy as the evaluation metric, which represents the average normalized accuracy for each experiment across all folds and for k=1 and k=3. Specifically, the following components are sequentially removed:

Learning a meta-vector to guide the update of classifier head module (meta-vector W): In this experiment, the one-dimensional vector responsible for learning the commonality across batch tasks was removed from the Eternal-MAML training strategy. However, the neural network model that uses either residual connections or the squeeze-and-excitation module was retained.

Squeeze-and-Excitation module (SE module) and Residual block: In this experiment, if both the SE module and the Residual block are removed from the neural network framework, a traditional Conv4 network is used as the framework for Eternal-MAML. If only one of these components is removed, ResNet-12 serves as the baseline model for the framework. Additionally, the Vector W is continuously learned in the outer loop to guide the updates of the classification head module during training phase. Table 5 presents the results of several ablation experiments conducted on the Eternal-MAML framework.

It can be seen from Table 5 that removing any component of Eternal-MAML results in a decrease in generalization performance. Specifically, when all components of Eternal-MAML are removed, its performance is the lowest, with a generalization accuracy of only 39.65%. When retaining the outer loop learning of W to guide the inner loop classification head module and using Conv4 as the neural network module, its average normalized accuracy is approximate 7% higher than all components are removed. It can indicate that introducing the W to guide the update of the inner loop classification head effectively addresses the label permutation problem in vanilla MAML. The conclusion is further validated in the next experiment, where the neural network module incorporates both the residual block and the squeeze-and-excitation module, but the label permutation method is unchanged, its generalization performance is not as good as that of the configuration containing only the vector W. Naturally, the experiment with the complete Eternal-MAML framework achieves the highest generalization accuracy, reaching 53.97%. Therefore, after these four experiments, it is evident that each main component of Eternal-MAML is indispensable. They interact with each other to efficiently adapt to cross-domain defect recognition tasks.

Table 5 Ablation study to identify the main components that contribute to the performance of Eternal-MAML.

Note that Meta-vector W represents a vector used to guide the weight updates of the classifier head. The significance of bold is the highlighting of the highest average normalized accuracies in the experimental results, which can infer that the Meta-vector W has the best performance. Best results are in bold.

Meta−vector W	SE module	Residual block	Accnor	
×	×	×	0.3965	
✓	×	×	0.4702	
×	✓	✓	0.4324	
✓	✓	✓	0.5397	

Impact of varying the count of inner loop steps

Extensive experiments are conducted in Ye & Chao (2021) to verify that MAML needs more inner loop update steps. The necessity arises because the model, once initialized and trained, must transition from making random predictions to achieving significantly higher classification accuracy. To verify whether increasing the number of inner loop update steps improves the performance of Eternal-MAML in the filed of cross-domain defect recognition, experiments were conducted on Eternal-MAML with the inner loop update steps m ∈ [1, 10] and ∂ ∈ [1e−4, 1e−1] during learning a 3-way classifier in fold 1, 2 and 3. As shown in Fig. 5, it can observe that as the number of inner loop steps increased, the model’s average normalized accuracy also increased accordingly. Specifically, the average normalized accuracy across three folds is only about 16% when the inner loop update step is set to As the number of steps increases, especially from 1 to 5, there is a notable improvement in Accnor, with an average increase of approximate 30% across all folds. However, when the update step count is increased to 10, the model’s generalization performance does not match that of the 5-step update, showing an average decrease of 1%. Furthermore, the computational cost for 10 update steps is twice that of five steps.

Figure 5 Experiments on three folds with the proposed framework reveal that within the inner loop update steps, the average normalized accuracy increases with the number of steps up to five.

However, when the steps are increased to 10, the performance does not surpass that of five steps. Additionally, the computational burden at 10 steps is significantly higher.

Although the viewpoint proposed in Raghu et al. (2019) does not apply to cross-domain defect recognition tasks, the experimental results on natural datasets indicate that increasing the number of inner loop update steps significantly improves generalization accuracy. Thus, we hypothesize that the number of inner loop update steps should be considered based on the specific type of task. Therefore, it can be inferred from Fig. 5 that setting m = 5 is the most appropriate choice for Eternal-MAML within the research problem of this article.

Comparsions of different network backbones

Raghu et al. (2019) and Tian et al. (2020) have verified from different perspectives a fundamental principle: the main idea of the few-shot classification problem is to train an excellent feature extractor. The number of layers in a neural network is proportional to the complexity of the task, so this experiment aims to verify whether the generalization performance improves correspondingly when the complexity of the neural network model used in the proposed framework increases. To this end, experiments to learn a 2-way classifier were performed on four different neural networks: Conv4, SE-ResNet-12, ResNet-18, and SE-ResNet-18. These neural network frameworks were applied to the Eternal-MAML, and experiments were conducted across all six folds. As shown in Table 6, the Conv4 backbone has the lowest parameter count and computational cost, yet it achieved an average normalized accuracy of 42.56%, which is higher than that of ResNet-18 and SE-ResNet-18. Although SE-ResNet-18 has more parameters than the SE-ResNet-12 used in the Eternal-MAML framework, and might theoretically extract more distinct features to enhance classification accuracy, the results indicate otherwise. The model with the SE-ResNet-12 backbone demonstrated the best generalization performance, achieving a normalized accuracy of 53.97%, which was approximate 12% higher than that of SE-ResNet-18. Thus, from the experiment, it can conclude that when selecting a backbone, it is essential to consider the complexity of the training task, and choosing an appropriate backbone can achieve ideal results in training.

Table 6 Comparing the generalization performance of Eternal-MAML on different neural network models.

It can be seen that when the Squeeze-and-Excitation module is added to ResNet-12 and applied to our proposed framework, it can extract key mutually exclusive features, achieving higher generalization accuracy in cross-domain defect recognition tasks. Best results are in bold.

Backbone	Params (M)	FLOPs (M)	Accnor	
Conv4	0.01	38.7	0.4256	
ResNet-18	26.2	555.6	0.4023	
SE-ResNet-18	26.5	556.3	0.4133	
ours(SE-ResNet-12)	12.6	159.9	0.5397	

Conclusion

In this article, a novel meta-learning framework, named as Eternal-MAML, is proposed, aiming at the cross-domain industrial defect recognition tasks. Eternal-MAML framework improves upon MAML by introducing a meta-vector that learns the commonality of batch tasks during training, and guides the update of the base learner’s classifier head, which addresses the label arrangement problem of vanilla MAML. Additionally, the original backbone network in MAML framework is modified by incorporating the Squeeze-and-Excitation module and Residual connection, resulting in more stable training and better generalization performance. In the simulation experiments, we rearranged the benchmark dataset MvTec to adapt to cross-domain defect recognition tasks. Experimental results show that Eternal-MAML outperforms the current state-of-the-art baseline models in all folds. Furthermore, ablation experiments were conducted on the main components of Eternal-MAML, and the results indicate that removing any part leads to a performance drop. However, the proposed framework exhibits limitations in terms of generalization accuracy in simulation experiments, indicating significant room for improvement. So in the future, we plan to improve at the feature extraction level by integrating advanced feature extraction methods for pixel-level defect recognition to enhance generalization accuracy during testing phase.

Supplemental Information

Supplemental Information 1 The source code of the model

Additional Information and Declarations

Competing Interests

Author Contributions

Data Availability

The authors declare there are no competing interests.

Jipeng Feng performed the experiments, performed the computation work, prepared figures and/or tables, and approved the final draft.

Haigang Zhang conceived and designed the experiments, performed the computation work, authored or reviewed drafts of the article, and approved the final draft.

Zhifeng Wang analyzed the data, authored or reviewed drafts of the article, and approved the final draft.

The following information was supplied regarding data availability:

The MVTec dataset is available at The MVTec anomaly detection dataset (MVTec AD): https://www.mvtec.com/company/research/datasets/mvtec-ad.

The NEU-CLS dataset is available at Zenodo:

Feng, J. (2025). Dataset [Data set]. In Yanqi Bao, Kechen Song, Jie Liu, Yanyan Wang, Yunhui Yan, Han Yu, Xingjie Li, ”Triplet- Graph Reasoning Network for Few-shot Metal Generic Surface Defect Segmentation,” IEEE Transactions on Instrumentation and Measuremente. Zenodo. https://doi.org/10.5281/zenodo.15098916.

The code is available at GitHub and Zenodo:

- https://github.com/zhg-SZPT/Eternal-MAML.

- Feng, J. (2025). code. Zenodo. https://doi.org/10.5281/zenodo.14799232.

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
