# Peer review of "Eternal-MAML: a meta-learning framework for cross-domain defect recognition"

_PeerJ Computer Science, doi:10.7717/peerj-cs.2757_

## Round 0.1 · original submission · Major Revisions

Please revise the work carefully according to the comments of the reviewers. Then your work will be evaluated again.

Reviewer 1 ·

Basic reporting

See below

Experimental design

See below

Validity of the findings

See below

Additional comments

This manuscript proposed the Eternal-MAML for the cross-domain industrial defect recognition tasks. However, during the review process, there are several serious problems that require improvement and clarification. The review comments are as follows:
1. It is suggested that the authors provide access to both the code and the associated dataset, as the lack of publicly available code and datasets poses significant challenges in verifying the authenticity of the method during the review process
2. It is suggested that the author further explain the failure cases and the shortcomings of the proposed method.
3. The comparative methods presented in this manuscript are not the most advanced and are not representative. It is suggested that the authors introduce at least five recently published few-shot defect classification methods for comparison.
4. In the part of comparative experiment, the author only described the advantages of the method, but did not analyze the reasons for its advantages.
5. It is recommended to validate on mainstream few-shot surface defect classification datasets such as NEU-CLS.
6. There are many references, but there are fewer references from 2023 to 2024. It is suggested that the authors cite more up-to-date references.
7. The experimental section lacks an introduction to the experimental environment and related hardware information.
8. Compared to traditional MAML algorithms, the innovation of this method is not obvious. The authors must highlight their research innovations. Additionally, the research motivation of the manuscript is not clear.
9. The author needs to explore more methods to elucidate the working mechanism and explainability of their model.

Cite this review as

Reviewer 2 ·

Basic reporting

This paper tackles the challenge of limited training samples in industrial defect detection by proposing Eternal-MAML, an enhanced MAML framework. It addresses label arrangement issues, integrates Squeeze-and-Excitation and Residual blocks for stability, and shows an 8% accuracy improvement over state-of-the-art methods on the MVTec dataset. Ablation studies further validate the framework's components. This work offers a promising solution for improving meta-learning models in industrial applications.

Experimental design

The experimental design is rational, covering the main needs of current industrial inspection while also demonstrating the superiority of the method through recognized evaluation criteria. The problem setup in the experiments is generally reasonable, reflecting the innovation and advantage of the method in this aspect. However, the description of the experimental methods and parameter settings should be more detailed, thoroughly describing the specific setup and corresponding results during the experimental process. Moreover, the lack of data description makes it difficult to ensure the reproducibility of the results. It is suggested to make the souce code public available on github.

Validity of the findings

The conclusions drawn in this paper effectively address the proposed issues and present viable solutions. Based on the experiments and their results, the method proposed in this paper exhibits good performance. However, the stability and generalizability of the method cannot be fully ascertained from the presentation in the paper. It is recommended to provide a more detailed description of the experimental data and an explanation of the corresponding experimental settings to enhance the persuasiveness and validity of the article.

Additional comments

The folloowing concerns should be addressed.
1.Clarification on the mechanism of vector W.
The manuscript mentions that the vector W improves the performance, but does not provide a clear explanation of how W guides the updates. I suggest adding a detailed description of the mechanism of W and its role in the model’s performance improvement. Providing a theoretical foundation or referencing similar studies would strengthen this section.
2.Unclear explanation in Lines 213-216?
The statement in lines 213-216 seems confusing and potentially misleading. It is unclear whether the issue arises from labeling of dataset, the MAML method, or a combination of both. I recommend revising this part to clarify the source of the problem. It would be helpful to provide more details and distinguish between these potential causes, supported by evidence from the text or relevant references.
3.Figure 1 does not explicitly show all modules.
The current version of Figure 1 does not clearly illustrate the SE module and residual connections. I suggest modifying the figure to explicitly depict these components or adding annotations that help the viewer intuitively understand the whole structure. This would enhance the clarity of the presentation.
4.Confusing definitions in Lines 268-274.
The definitions of “transfer learning” and “learning from scratch” presented in lines 268-274 are not entirely clear. Specifically, it is unusual to define transfer learning (TF) as training with a larger train set, and learning from scratch (LFS) as training with a smaller test set. Additionally, the explanation that LFS is initialized using pre-trained weights from ImageNet does not align with the traditional method of training from scratch. I suggest revising these definitions to align with standard usage in the literature and providing clearer justification for the choices made in the study.

Cite this review as

---

## Round 0.2 · accepted · Accept

Thanks a lot for your efforts to improve the work. This version successfully satisfied the reviewer, and I believe it can be accepted now. Congrats!

Reviewer 1 ·

Basic reporting

0

Experimental design

0

Validity of the findings

0

Cite this review as